# Non-pharmacological fatigue interventions for patients with a primary brain tumour: a scoping review protocol

Rachael Simms-Moore [1], Emma Dures [1], Neil Barua,[2] Fiona Cramp [1]

¹School of Health and Social Wellbeing, University of the West of England, Bristol, UK
²Neurosurgery, Southmead Hospital, Bristol, UK

**Correspondence to**
Ms Rachael Simms-Moore; rachael.simms-moore@uwe.ac.uk

## ABSTRACT

**Introduction** Fatigue is the most prevalent symptom for patients with a primary brain tumour (PBT), significantly reducing quality of life and limiting daily activities. Currently, there are limited options for managing cancer-related fatigue (CRF) in patients with a PBT, using non-pharmacological methods. The objective of this scoping review is to identify current and emerging evidence in relation to non-pharmacological CRF interventions for patients with a PBT.

**Methods and analysis** Electronic databases OVID and EBSCO platforms: MEDLINE, EMBASE and CINAHL will be searched. In addition, PROSPERO, The Cochrane Library and ISI Web of Science will be searched. Trials registries CENTRAL and the International Clinical Trials Registry platform will also be searched for ongoing research. Inclusion criteria: studies from 2006 onwards, primary research on non-pharmacological interventions in patients with a PBT (>18 years). A Preferred Reporting Items for Systematic Reviews and Meta-Analyses (PRISMA) flow diagram will be utilised to summarise the screening process and results.

Quantitative data will be analysed descriptively, while content analysis will be used for qualitative data. Findings will map the existing and emerging evidence on non-pharmacological interventions for CRF in patients with PBTs. This will provide insights into the extent and nature of the evidence in this evolving field, identifying gaps in knowledge and research priorities, and guide further investigations in this area.

**Ethics and dissemination** Ethical approval is not required for this scoping review. Findings will be disseminated via relevant peer-reviewed journals, PhD thesis, conference presentations, and shared with relevant charities and health professionals.

## INTRODUCTION

A primary brain tumour (PBT) is a growth of cells originating in the brain that multiplies in 'an abnormal, uncontrollable way'.[1] PBTs can be categorised into high grade (grades 3–4) and lower grade (grade 1–2) tumours, with high-grade PBTs exhibiting more rapid growth than lower-grade PBTs.[1] Gliomas are the most common type of PBT, arising from

---

**STRENGTHS AND LIMITATIONS OF THIS STUDY**

⇒ Using the Preferred Reporting Items for Systematic Reviews and Meta-Analyses Extension for Scoping Reviews framework ensures replicability and transparency.
⇒ Two independent researchers will assess the included studies, enhancing the reliability of the review.
⇒ Inclusion of studies from 2006 onwards captures recent and relevant research.
⇒ Broad evidence capture addresses an under-researched problem in a niche patient population.
⇒ Potential bias due to the exclusion of studies not available in English.

---

glial cells in the brain and accounting for over half of PBTs.[2]

Fatigue is the most prevalent symptom for patients with PBT.[3] Fatigue associated with having cancer and consequent treatments such as radiotherapy and chemotherapy is known as cancer-related fatigue (CRF).[4] The National Comprehensive Cancer Network defines CRF as 'a distressing, persistent, subjective sense of physical, emotional, and/or cognitive tiredness or exhaustion related to cancer or cancer treatment that is not proportional to recent activity and interferes with usual functioning'.[5] CRF can involve mental, physical and/or psychological components, with variability between patients and over time.[6] Along with experiencing unusually elevated levels of fatigue disproportionate to activity, CRF cannot be mitigated through sleep or rest.[7] CRF can significantly reduce cancer patients' quality of life,[8] limiting their ability to work, continue treatment, socialise or engage in daily activities.[9–11] Indeed, a study found that patients felt that CRF negatively impacted daily life more than pain.[4]

PBTs pose distinct challenges in the management of CRF due to their direct impact on brain functions that regulate energy, sleep, mood and cognition.[12] This

complexity is exacerbated by the diverse manifestations of fatigue, which arise not only from the tumour's metabolic demands and its interference with neural pathways but also from the array of treatments such as surgery, chemotherapy and radiation.[12] The potential use of anti-epileptic drugs (AEDs), which are commonly prescribed for PBT-induced seizures but are rare in the treatment of other cancer types can also contribute to fatigue.[13] The trajectory of PBTs can vary significantly; high-grade tumours may progress rapidly, demanding immediate management of symptoms like CRF, while low-grade tumours may evolve more slowly, allowing for a more measured approach.[1] This suggests a need for tailoring of support to the patient journey.

There are two potential avenues for treating CRF, pharmacological interventions and non-pharmacological interventions. Non-pharmacological interventions are defined as interventions where pharmaceutical medications are not used as treatment,[14] for example, lifestyle changes, psychosocial therapies and physical activity.

Evidence suggests that many patients do not want to take more drugs, with approximately 50% of patients with chronic illness not taking their medications as prescribed.[15 16] A Cochrane review (updated 2022) of 'Interventions for the management of CRF in adults with a primary brain tumour' explored efficacy of interventions. This review included three randomised control trials (RCT) that investigated pharmacological interventions for CRF.[17] All three medicines studied, modafinil, dexamfetamine sulphate and amodofinil are psychostimulants; based on the theory that promoting wakefulness could reduce CRF in patients with PBT. All three studies reported no significant effect of the pharmacological intervention on CRF compared with the control arm.[18–20] Currently, there are limited options for managing CRF in patients with a PBT.[17] Non-pharmacological evidence to date has predominantly focused on movement interventions, physical activity and exercise. Although the terms have been used interchangeably,[21] they differ. Exercise interventions being structured, planned, repetitive activities aimed at improving fitness and health outcomes and physical activity interventions covering a broader range of movement aiming to increase overall activity levels for health and well-being.[22] In a systematic review of physical activity and exercise in adults with primary brain cancer, two RCTs reported improvements in patient fatigue.[23]

One pilot RCT focused on an exercise programme to improve cognitive function in patients with a PBT. It demonstrated small to medium self-reported improvements in CRF, specifically in the areas of physical fatigue and reduced activity, with a medium effect compared with the control arm.[24]

Another study by Milbury et al[25] explored the effectiveness of dyadic yoga as an intervention for glioma patients and their families. Their study demonstrated a small clinically significant difference between the control and intervention groups, favouring the dyadic yoga intervention.[25]

Overall, these findings highlight the potential of physical activity interventions in reducing CRF in patients with a PBT.

There is some limited research indicating that patients with PBT may benefit from cognitive rehabilitation programmes[23] or health coaching programmes.[26] Although not specifically studied in patients with a PBT, cognitive behavioural therapy (CBT) has been shown to reduce severe CRF in a variety of cancers.[27 28] CBT is a talking therapy that helps treat mental health problems by focusing on how thoughts, beliefs and attitudes impact emotions and behaviours.[29] CBT teaches coping skills and challenges negative thinking patterns, aiming to improve emotional well-being.[29]

Within the Cochrane review, non-pharmacological studies were organised into four main categories: educational programmes, physical activity, psychosocial/talking therapies and cognitive rehabilitation programmes. Overall findings indicated that there was insufficient evidence to support the effectiveness and safety of non-pharmacological treatments for CRF in patients with PBT.[17] In contrast, while pharmacological interventions have shown limited benefits in addressing CRF and patients often express hesitation in adding more medications to their regimen, emerging evidence beyond the scope of the Cochrane review suggests some benefits of non-pharmacological interventions in the management of CRF.

However, the RCT evidence specific to PBT is limited. Considering primary evidence, beyond RCTs has the potential to capture a wider range of interventions, contributing to a better understanding of current knowledge in this area that could benefit patients with PBT.

## Review aims

This review will focus on non-pharmacological interventions for managing CRF in patients with a PBT to provide a comprehensive map of published and grey literature, with the aim of highlighting knowledge gaps and identifying priority areas for future research.

## METHODS AND ANALYSIS
### Study design

Scoping reviews serve as an invaluable tool when delving into emerging research areas.[30] Unlike systematic or Cochrane reviews, which often focus on specific research designs like RCTs, scoping reviews cover a broader range of methods, ensuring a more comprehensive overview of the topic.[17 31] This is beneficial when navigating evolving fields, such as non-pharmacological interventions for CRF in PBT patients. Our scoping review aims to bridge this gap, offering readers a comprehensive view of the current research landscape, highlighting both its strengths and areas of opportunity for further investigation.

To ensure a comprehensive and systematic approach to our scoping review, we will be employing the methodology as delineated by Arksey and O'Malley and further refined

by Levac *et al*.[31] This methodology is widely recognised for its structured and transparent nature, and it encompasses the following stages:

Defining the research question: we have established a clear aim for this review, ensuring that our objectives are well-defined and aligned with the scope of our investigation.

Identification of relevant studies: we will implement a rigorous and comprehensive search strategy across multiple databases. This will ensure that we capture a broad spectrum of literature pertinent to our research question.

Selection of studies: based on our pre-established inclusion and exclusion criteria, we will systematically select studies that align with the objectives of our review.

Data charting: we will extract essential details and findings from each selected study. This step will be carried out with consistency to ensure uniformity in the data collection process.

Analysis and reporting: once the data is charted, we will analyse it to discern patterns, emerging themes, and any evident gaps in the literature. This will provide a holistic view of the current research landscape.

Stakeholder consultation: while this step is optional, we are engaging with relevant stakeholders. Their insights and perspectives can offer additional depth to our findings and ensure that our review remains relevant to the broader community.

By adhering to this structured methodology, we aim to produce a scoping review that offers a comprehensive, transparent and insightful overview of the existing research on our topic.

This review will use the Preferred Reporting Items for Systematic Reviews and Meta-Analyses (PRISMA) Extension for Scoping Reviews framework (see online supplemental information research checklist).[30]

### Study question

The review will address the question: what interventions have addressed the management of CRF in patients with PBT? The review aims to map the evidence, identify the range, recognise gaps and highlight emerging trends. By understanding the current landscape, the review will inform future research priorities.

### Identifying relevant studies

The scoping review will include all primary research designs published from 2006 onwards. This date corresponds with the National Institute for Health and Clinical Excellence (NICE) introducing updated guidance 'Improving Outcomes for People with Brain and Other CNS Tumours'.[32] This guidance marked a significant shift in the approach to care for PBT patients; promoting multidisciplinary care throughout the cancer journey to reflect fluctuating need.[33] NICE guidance is likely to have highlighted a gap in research evidence for the management of CRF in patients with PBT, although the first RCT specific to this was not published until 2013.[18] Commencing the

scoping review search prior to publication of the updated NICE guidance is not likely to identify relevant studies.

A search strategy has been developed with input from a university-based specialist subject librarian and modified for each database (see online supplemental information 1–8). The Peer Review of Electronic Search Strategy checklist will be used to validate the search strategy.[34]

To ensure a comprehensive literature search, a range of generalised and specialised electronic databases will be searched.[35 36] Details of the keyword searches are presented in online supplemental information 1–8. The following electronic databases will be searched using the OVID and EBSCO platforms: MEDLINE, EMBASE and CINAHL. In addition, PROSPERO, The Cochrane Library and ISI Web of Science will be searched. Trials registries CENTRAL and the International Clinical Trials Registry platform will also be searched for ongoing research.

The databases will be searched using Medical Subject Headings (MeSH) and key search terms (see online supplemental information 9). Keywords for each subject heading were developed using a mix of the Yale MeSH analyser software,[36] free-text terms and explode command in MEDLINE, thesaurus recommendations and preferred vocabulary across databases.

In-process research such as MEDLINE—IN PROCESS and other non-indexed citations will be searched via OVID to minimise publication bias. Grey literature database Ethos will also be searched to minimise bias as positive findings are more likely to be published within peer-reviewed journals.[37] A PICO model demonstrates the concept and context of this search strategy (see online supplemental information 10). The lead author (RS-M) will upload all studies and articles into referencing software EndNote-20[38] and use scoping review software Covidence (Veritas Health Innovation, Melbourne, Australia) to store the data. Any duplicates will be removed at this stage. To identify additional relevant studies that might have been missed in the initial database search the reference lists of included papers will be searched. For grey literature, we will employ the de-duplication strategy as suggested by Bramer *et al*.[39]

### Selection of studies

Studies found will then be screened using inclusion and exclusion criteria (table 1).

In this review, samples of adult patients with PBTs of grade 2–4 will be included. This is supported by previous research by Röttgering[3], finding that there was not a significant difference in fatigue severity reported by individuals with different tumour grades ranging from 2 to 4.[3]

To ensure a holistic representation of the non-pharmacological strategies employed for PBT patients, it is imperative to encompass studies that combine multiple interventions. This approach not only broadens the scope of our review but also mirrors the multifaceted care strategies often employed in clinical settings.

The screening process will begin with RS-M and another independent reviewer, reviewing the titles, followed by

**Table 1** Inclusion and exclusion criteria for studies

| Inclusion | Exclusion |
|---|---|
| ► Any primary research designs<br>► Published from year 2006 onwards<br>► Available in English language<br>► Patients with a PBT ≥18 years<br>► Includes separate data for adults with a PBT<br>► PBT Glioma grades 2–4<br>► Studies including mixed non-pharmacological interventions<br>► Non-pharmacological intervention being investigated for its effect on CRF | ► Animal studies<br>► Children (persons under 18 years)<br>► Editorial or commentaries<br>► Secondary data such as systematic reviews<br>► Studies including participants with grade 1 or metastatic brain tumours<br>► Childhood brain tumours in patients who are now adults |

CRF, cancer-related fatigue; PBT, primary brain tumour.

the abstracts, and any resources with doubt or disagreement will be retained. Next, both reviewers will independently screen the full texts using the inclusion and exclusion criteria. If there are any disagreements these will be discussed initially and if necessary, the director of studies (FC) will be consulted for conciliation.

### Data charting process
Study characteristics, design, aims, participants, interventions, outcome measures, findings and authors conclusions will be collected into a data charting form (DCF) (see online supplemental information 11). The DCF will initially be piloted by the lead author and director of studies on 4–5 papers (RS-M and FC) and refined as necessary. An independent reviewer will carry out data extraction on the studies for process validation.

### Analysis of data
Data analysis is likely to be narrative with content analysis used for qualitative data. All data will be summarised to produce a map of the current literature available.

Content analysis will be conducted following a systematic, inductive approach.[40] First, we will perform data collection by extracting key information from each study using a standardised form into the DCF. Next, we will complete coding, where specific data pieces will be labelled to categorise them; for example, we might use the label 'Physical Interventions' for all strategies related to physical activity. After coding, we will proceed to Themes, where we group similar codes together to identify broader themes, such as grouping various interventions under 'Lifestyle Interventions'. This will be followed by refinement, where we will adjust categories to ensure clarity and accuracy in our analysis. Subsequently, we will move to interpretation, where we will analyse the data to identify patterns and gaps in the literature. To present our findings effectively, we will employ visualisation techniques, including tables or charts. Finally, to ensure the reliability of our analysis, multiple reviewers will code some studies, and any discrepancies will be resolved for consistency. Through this comprehensive inductive content analysis approach, we aim to systematically categorise and interpret the literature, shedding light on key trends and areas that require further research.

### Assessment of methodological quality of individual studies
Two reviewers (RS-M and one other) will independently complete a quality assessment of each included study using the enhanced Mixed Methods Appraisal Tool (MMAT). The MMAT was selected as the review is likely to contain quantitative and qualitative studies, therefore a mixed methods quality assessment tool is appropriate.[41] The enhanced MMAT version was selected as it focuses on five core quality criteria to increase efficiency.[42]

### Collating, summarising and reporting
Gaps will be highlighted and where appropriate recommendations for future research and clinical practice will be identified. Findings will be presented in a PRISMA flow diagram, mapping the number of resources identified and reasons for those included and excluded.[30] Included studies will be summarised in tabular format. Further figures may be used to illustrate a map of the literature and gaps highlighted.

### Patient and public involvement
This research has been discussed with patient research partners and representatives of two relevant brain tumour charities who agree that this is a priority for research. They will continue to work alongside the lead researcher in an advisory role and the findings from the review will be shared with them for further discussion. The findings derived from this study have the potential to inform and guide the research team in their endeavour to design a non-pharmacological intervention for managing fatigue in patients with a brain tumour.

### DISCUSSION
There is no universally accepted standard treatment for CRF, and a broad range of non-pharmacological strategies exist for managing fatigue in patients with a PBT.[17] This scoping review aims to expand on the findings of a previous Cochrane review (2015) and comprehensively examine the available evidence encompassing a broad range of non-pharmacological interventions for CRF in patients with PBT. By incorporating all primary research evidence, the review seeks to provide a comprehensive overview of the extent and type of evidence available in this developing field. It aims to

identify gaps and priorities for future research, ultimately contributing to a better understanding of the current state of knowledge in this area. The results may identify interventions that are worthy of further research investment and provide recommendations for clinical practice. Through this exploration of non-pharmacological interventions for managing CRF in patients with PBT, we aim to enhance understanding and guide decision-making and identify potential interventions that could benefit patients with a PBT in the future.

## Ethics and dissemination

Ethical approval is not required for this scoping review. The findings of this review will be disseminated via relevant peer-reviewed journals, PhD thesis, conference presentations and through sharing findings with relevant charities and health professionals.

## Amendments

The protocol will be closely followed throughout with RS-M regularly reporting to the supervisory team (FC, NB and ED). If any amendments are made to the published study protocol, these will be reported in the final publication.

**Contributors** RS-M and FC proposed and developed the scoping review protocol topic. Ideas were developed further with academic supervisors NB and ED. The academic supervisors read and approved the final manuscript.

**Funding** This scoping review is funded as part of a PhD studentship by the University of the West of England Bristol and Somerset, Wiltshire, Avon and Gloucestershire (SWAG) Cancer Services held by RS-M, Project 9187471. The funders had no role in this scoping review protocol.

**Competing interests** None declared.

**Patient and public involvement** Patients and/or the public were involved in the design, or conduct, or reporting, or dissemination plans of this research. Refer to the Methods section for further details.

**Patient consent for publication** Not applicable.

**Provenance and peer review** Not commissioned; externally peer reviewed.

**ORCID iDs**
Rachael Simms-Moore http://orcid.org/0009-0003-7383-0355
Emma Dures http://orcid.org/0000-0002-6674-8607
Fiona Cramp http://orcid.org/0000-0001-8035-9758

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
