## [Reviewer comments · BMJ Open]

ARTICLE DETAILS

TITLE (PROVISIONAL)	Non-pharmacological fatigue interventions for patients with a primary brain tumour: a scoping review protocol.
AUTHORS	Simms-Moore, Rachael; Dures, Emma; Barua, Neil; Cramp, Fiona

VERSION 1 – REVIEW

REVIEWER	Guldager, Rikke Copenhagen University Hospital, Department of Neurosurgery
REVIEW RETURNED	10-Aug-2023

GENERAL COMMENTS	Thank you for letting me review your scoping review protocol on an exciting topic. Unfortunately, I do not believe that the rationale for conducting the study is strong enough and I do not think that the chosen method is appropriate. Therefore, I have to reject the paper. I have provided some comments Abstract A structured summary has been provided Strengths and limitations of the study: It seems like a limitation that you only include literature in English. I am not quite sure if it is a limitation or a strength that studies from 2006 is included. Please provide an argument Ethics approval and informed consent Overall comment: There is different font on the paper Introduction: The rationale for doing a ScR is not clear. With this aim it would be more appropriate to conduct a systematic review. Unclear why PBT is different from other cancer populations and how this affects CRF. Further, it is unclear whether it is PBT that gives symptoms or CRF (page 3, lines 16-18). I suggest that the aspect of time is mentioned since PBT is a rapid and life-threatening disease. How does that affect the intervention provided to this patient population? P. 3, Line 34 Not abbreviation here P 3, Line 49 I suggest removing 'recent' Methods and analysis Study design: A scoping review aims to systematically identify and map the breadth of evidence on a particular topic and does not usually make a formal quality appraisal. Thus, choosing to conduct a ScR is unclear. You refer to a Cochran review from 2022 (Interventions for the management of Cancer-related fatigue in adults with a primary brain tumor) and it is unclear why you want to do an ScR when there have already been done some work in this area.
--

	Prisma-ScR can be used for reporting scoping reviews, but what methodology will be used? That is unclear Study Question: To evaluate the quality of evidence is as written above not within the remit of a scoping review. Thus, the research question and objectives of the study do not lend to the scoping review approach. Further, the research questions seem too specific for a scoping review. Identifying relevant studies: P 5, line 26 no need to refer to Covidence. Instead, write... Covidence (Veritas Health Innovation, Melbourne, Australia) to store the data. Table 1: Inclusion criteria: ≥18 years Studies including mixed non-pharmacological intervention? Generally, on inclusion: Why studies from 2006? Exclusion criteria: Childhood brain tumors in patients who are now adults Search process: I suggest that you make a PICO model (separate it from suppl 2), so the concept and context are clearer. I suggest that you add the following search terms to concept 4: measure, measurements, scale, education, information, instruments Are there any limitations to the search besides the year? Will the reference list of pertinent papers be searched? All information e.g., databases with dates are missing and should be provided Data collection: How will you manage gray literature? I suggest that you refer to Bramer, W.M., et al., De-duplication of database search results for systematic reviews in EndNote. J Med Libr Assoc, 2016. 104(3): p. 240-3. Data charting process: How many papers will the DCF be piloted on? Study location, is that setting or country? Could it be interesting to collect data on who is providing the intervention? Analysis: Please elaborate on content analysis. How will this be performed? More details are needed on this. Discussion The dates of the study are missing References: Please make sure that all the journals are correctly abbreviated
--	---

REVIEWER	Day, Julia NHS Lothian
REVIEW RETURNED	30-Aug-2023

GENERAL COMMENTS	I hope the scoping review will be a useful addition to the Cochrane systematic review on the same topic. It would be helpful to have further justification as to why the year 2006 was identified as the starting point for the search, for example referring to a systematic review associated with the NICE guidelines that may have captured research prior to this year.
--

VERSION 1 – AUTHOR RESPONSE

Reviewer's comments:

Reviewer #1:

1. Thank you for letting me review your scoping review protocol on an exciting topic. Unfortunately, I do not believe that the rationale for conducting the study is strong enough and I do not think that the chosen method is appropriate. Therefore, I have to reject the paper.

Response:

We appreciate your feedback and have addressed the concerns raised. We have reworked and strengthened the rationale for our study and provided more justification for the chosen scoping review method in the introduction and methods sections of the revised manuscript.

2. It seems like a limitation that you only include literature in English. I am not quite sure if it is a limitation or a strength that studies from 2006 are included. Please provide an argument.

Response:

We have revised the 'Strengths and limitations of this study' section and reworded the point on including studies from 2006 and hope that it is now clearer that this is a strength. Further detail has already been presented in the main methods to justify this date.

The revised text reads as follows:

- Using the PRISMA- Extension for Scoping Reviews framework ensures replicability and transparency.
- Two independent researchers will assess the included studies, enhancing the reliability of the review.
- Inclusion of studies from 2006 onwards captures recent and relevant research.
- Broad evidence capture addresses an under-researched problem in a niche patient population.
- Potential bias due to the exclusion of studies not available in English.

[page 2, Lines 35-42].

3. Unclear why PBT is different from other cancer populations and how this affects CRF. Further, it is unclear whether it is PBT that gives symptoms or CRF (page 3, lines 16-18). I suggest that the aspect of time is mentioned since PBT is a rapid and life-threatening disease. How does that affect the intervention provided to this patient population?

Response:

We have provided a clearer explanation of why PBT is distinct from other cancer populations and how this distinction relates to CRF. We have also addressed the aspect of time and its impact on interventions for PBT patients in the revised manuscript. The revised text reads as follows,

“PBTs pose distinct challenges in the management of (CRF) due to their direct impact on brain functions that regulate energy, sleep, mood, and cognition [12] This complexity is exacerbated by the diverse manifestations of fatigue, which arise not only from the tumour's metabolic demands and its interference with neural pathways but also from the array of treatments such as surgery, chemotherapy, and radiation [12]. The potential use of neuro-epileptics, which are commonly prescribed for PBT-induced seizures but are rare in the treatment of other cancer types can also contribute to fatigue [13]. The trajectory of PBTs can vary significantly; high-grade tumours may progress rapidly, demanding immediate management of symptoms like CRF, while low-grade tumours may evolve more slowly, allowing for a more measured approach [1]. This suggests a need for tailoring of support to the patient journey.”

[page 3, Lines 19-29]

4. P. 3, Line 34: No abbreviation here.

Response:

We have now included the full form “randomised controlled trials along with its abbreviation RCT.

[page 3, Lines 38]

5. P 3, Line 49: I suggest removing ‘recent.’

Response:

We have removed the word "recent" as recommended.

[page 3, Line 35]

6. A scoping review aims to systematically identify and map the breadth of evidence on a particular topic and does not usually make a formal quality appraisal. Thus, choosing to conduct a ScR is unclear. You refer to a Cochran review from 2022 (Interventions for the management of Cancer-related fatigue in adults with a primary brain tumour) and it is unclear why you want to do an ScR when there have already been done some work in this area.

Response:

We have revised our review aims for clarity showing quality appraisal is not in the scope of this review. The revised text reads as follows,

“This review will focus on non-pharmacological interventions for managing CRF in patients with a PBT to provide a comprehensive map of published and grey literature, with the aim of highlighting knowledge gaps and identifying priority areas for future research.”

[page 4, Lines 32-34]

We have also provided a clearer justification for choosing a scoping review approach and address why it is relevant, especially considering existing systematic reviews.

[page 4, Lines 38-44]

7 Prisma-ScR can be used for reporting scoping reviews, but what methodology will be used? That is unclear.

Response:

We have provided further clarity in the revised manuscript. Specifically, we are employing the methodology for scoping reviews as outlined by Arksey and O'Malley, which has been further refined by Levac et al. This methodology is recognized for its systematic and transparent approach, and it encompasses the following stages:

- Research Question: The primary aim and objectives of the review are clearly defined.
- Identify Studies: A comprehensive search strategy is employed across multiple databases to ensure a broad capture of relevant literature.
- Study Selection: Studies are selected based on predetermined inclusion and exclusion criteria.
- Chart Data: Key details and findings are extracted from the selected studies, ensuring a consistent approach to data collection.
- Analyse & Report: The collated data is then analysed to identify patterns, themes, and potential gaps in the existing literature.
- Consultation: As an optional step, we may engage with stakeholders to gain additional insights and perspectives on the findings.

By adhering to this methodology, we aim to provide a thorough and transparent overview of the current state of research on the topic.

[page 4, Lines 45-48; page 5, Lines 1-28]

8. To evaluate the quality of evidence is as written above not within the remit of a scoping review. Thus, the research question and objectives of the study do not lend to the scoping review approach. Further, the research questions seem too specific for a scoping review.

Response:

Thank you for your feedback. We have reframed the research questions and objectives to align better with the scoping review approach and ensure they are less specific. We have also emphasised that the purpose of the scoping review is not to evaluate the quality of evidence.

[page 4, Lines 32-34] [page 5, Lines 30-33]

9. P 5, Line 26: No need to refer to Covidence. Instead, write... Covidence (Veritas Health Innovation, Melbourne, Australia) to store the data.

Response:

We have made the suggested change to the reference to Covidence in the text and removed it from our reference list.

[page 6, Line 15]

10. Data collection: How will you manage gray literature? I suggest that you refer to Bramer, W.M., et al., De-duplication of database search results for systematic reviews in EndNote. J Med Libr Assoc, 2016. 104(3): p. 240-3.

Response:

We have added to the Data charting process that for gray literature, we will employ the de-duplication strategy as suggested by Bramer et al. (2016).

[page 6, Lines 16-19]

11 Analysis: Please elaborate on content analysis. How will this be performed? More details are needed on this.

Response:

We have provided a more detailed description of how content analysis will be performed in the revised manuscript. The revised text reads as follows,

"Content Analysis will be conducted following a systematic, inductive approach. First, we will perform Data Collection by extracting key information from each study using a standardised form. Next, we will complete Coding, where specific data pieces will be labelled to categorise them; for example, we might use the label "Physical Interventions" for all strategies related to physical activity. After coding, we will proceed to Themes, where we group similar codes together to identify broader themes, such as grouping various interventions under "Lifestyle

Interventions." This will be followed by Refinement, where we will adjust categories to ensure clarity and accuracy in our analysis. Subsequently, we will move to Interpretation follows, where we will analyse the data to identify patterns and gaps in the literature. To present our findings effectively, we will employ Visualisation techniques, including tables or charts. Lastly, to ensure the reliability of our analysis, multiple reviewers will code some studies, and any discrepancies will be resolved for consistency. Through this comprehensive inductive content analysis approach, we aim to systematically categorise and interpret the literature, shedding light on key trends and areas that require further research."

[page 7, Lines 9-22]

12 References: Please make sure that all the journals are correctly abbreviated.

Response:

We have ensured all Journal abbreviations have been corrected.

[page 9-11]

13 Discussion: The dates of the study are missing.

Response:

We have ensured that the Cochrane study date is now included in the revised manuscript.

[page 7, Line 45]

14 Table 1: Inclusion criteria:
≥18 years

Response:

We have now included "≥18 years" in our Inclusion column in Table 1.

[pages 6, Table 1]

15 Table 1: Inclusion criteria:
Studies including mixed non-pharmacological intervention?

We have chosen to include studies including mixed non-pharmacological intervention as this should capture the maximum number of non-pharmacological interventions across the literature. We have justified this further in the manuscript. The revised text reads as follows,

We have chosen to include studies featuring mixed non-pharmacological interventions to ensure a comprehensive capture of the diverse non-pharmacological approaches that have been investigated. We provide further justification for this decision in the manuscript. The revised text reads as follows,

“To ensure a holistic representation of the non-pharmacological strategies employed for PBT patients, it is imperative to encompass studies that combine multiple interventions. This approach not only broadens the scope of our review but also mirrors the multifaceted care strategies often employed in clinical settings.”

[page 6, Lines 27-30]

16 Table 1: Inclusion criteria:

Generally, on inclusion: Why studies from 2006?

We have included studies from 2006 onward in accordance with the introduction of the NICE guidelines. This justification is presented in the manuscript.

[page 5, Lines 35-43]

17 Table 1: Exclusion criteria:

Childhood brain tumours in patients who are now adults

Response:

Thank you, we have now included the additional exclusion criteria “childhood brain tumours in patients who are now adults” in table 1 as per your suggestion.

[page 6, Table 1]

18 Search process:

I suggest that you make a PICO model (separate it from suppl 2), so the concept and context are clearer.

Response:

As per your recommendation, we have developed a PICO model and separated it from previously names Supplement 2: Scoping Review Search Strategy overview. The PICO model is now named Supplement 10 and the Supplement 2: Scoping Review Search Strategy overview has been named Supplement 9 with the inclusion and exclusion criteria removed for clarity. We hope this will aid in better defining our research concepts and context, ensuring a more systematic and focused review process. However, we also note that in a scoping review the comparison element is not explicitly defined. We have included the line “Not explicitly defined in scoping reviews” for the PICO model comparison arm.

[Supplement 9-10]

19 I suggest that you add the following search terms to concept 4: measure, measurements, scale, education, information, instruments.

Response:

In response to your comment, we have incorporated the recommended terms "measure," "measurements," "scale," "education," "information," and "instruments" to concept 4 of our search strategy overview and all database search strategies.

[Supplement 1-9]

20 Are there any limitations to the search besides the year?

Response:

There is a potential bias due to the exclusion of studies not available in English. This has now been included as a limitation in the strengths and limitations article summary.

21 Will the reference list of pertinent papers be searched?

Response:

Yes, the reference lists of pertinent papers will be searched in our scoping review. We have now included this point in the protocol methodology. The revised text reads as follows,

“To identify additional relevant studies that might have been missed in the initial database search the reference lists of included papers will be searched.”

[page 6, Lines 16-18]

22 All information e.g., databases with dates are missing and should be provided.

Response:

Thank you for the comment. We acknowledge the importance of specifying the databases and their respective search dates. As the scoping review is still in the planning stages, we have not yet finalised these details. The specific databases and search dates will be comprehensively detailed in the final scoping review manuscript.

23 Data charting process:

How many papers will the DCF be piloted on?

Response:

To ensure its comprehensiveness and effectiveness. For this review, we plan to pilot the DCF on 4-5 papers. This approach allows us to identify and address any challenges or

inconsistencies in data extraction early on, ensuring a smooth and consistent data charting process for all included papers. We have now added this clarification into the text.

[page 7, Lines 3-5]

24 Study location, is that setting or country?

Response:

We have now included study location and study setting for clarity in the data charting form. "Study location" refers to the country or geographic region where the study was conducted. The "setting" refers to the specific environment or context within that location where the study took place, such as a hospital, community centre, charity centre, or home. So, "Study location" is the country, while "setting" will provide more detailed information about where within that country the study occurred.

[Supplement 9]

25 Could it be interesting to collect data on who is providing the intervention?

Response:

Thank you for this comment. We recognise the importance of understanding who is providing the intervention, as it can offer valuable context and potentially influence the outcomes of the intervention. In response to your suggestion, we have now incorporated additional fields in the Data Charting Form (DCF) to systematically capture this aspect.

Specifically, we have added a dedicated field titled "Intervention provider" in the DCF. This will include pre-defined categories such as medical professionals, allied health professionals, laypersons or trained volunteers, patients' family members or caregivers, and an "Others" category with space to specify unique provider types. Additionally, we've made provisions to capture any specific training, certification, or qualifications the provider might have that's pertinent to the intervention in a additional field titled "Intervention provider training".

We believe that by capturing this information, we can offer a more comprehensive understanding of the role of intervention providers in the studies we're reviewing. Once again, thank you for highlighting this important aspect.

[Supplement 11]

Reviewer #2:

1. I hope the scoping review will be a useful addition to the Cochrane systematic review on the same

topic. It would be helpful to have further justification as to why the year 2006 was identified as the starting point for the search, for example referring to a systematic review associated with the NICE guidelines that may have captured research prior to this year.

RESPONSE

We have included studies from 2006 onward in accordance with the introduction of the NICE guidelines. Further justification is presented in the manuscript.

[page 5, Lines 35-43]